# Indoor Air Quality Sensor Utilization for Unwanted Fire Alarm Improvement in Studio-Type Apartments

**Han-bit Choi** **, Euy-hong Hwang and Don-mook Choi ***

Department of Equipment System and Fire Protection Engineering, Gachon University, 1342, Seongnam-daero, Sujeong-gu, Seongnam-si 13120, Republic of Korea; hanbisi970@gachon.ac.kr (H.-b.C.); dmlghd2@gachon.ac.kr (E.-h.H.)
* Correspondence: fire@gachon.ac.kr

**Abstract:** Smoke detectors play a vital role in evacuation and safety during fire incidents, as they directly contribute to the reliability and accuracy of firefighting systems. However, if not installed properly, smoke detectors can trigger unwanted fire alarms (UWFAs), particularly in studio-type apartments. Therefore, this study aimed to develop a method for reducing UWFAs by addressing the challenges posed by cooking by-products in such environments. The proposed algorithm was validated through tests, considering relevant literature and standards, and utilizing indoor air quality sensors. Verification tests were conducted to enhance the accuracy of the algorithm. Based on the experimental results, cutoff values of 5 ppm for CO and 7000 $\mu g/m^3$ for PM10.0 were proposed as criteria for identifying UWFAs caused by cooking by-products.

**Keywords:** unwanted fire alarm; studio-type apartment; cooking nuisance; indoor air quality sensor; thin-sliced pork belly; mackerel; smoke detector; carbon monoxide; particulate matter

## 1. Introduction

### 1.1. Research Background

The smoke detector is a device that is sensitive to particulate combustion products and/or thermal decomposition suspended in the atmosphere [1]. It is a firefighting product that works in the early stage of a fire to help evacuation and extinguishment. Thus, smoke detectors are directly involved in evacuation and safety in the event of a fire. They also significantly affect the reliability and accuracy of firefighting facilities [2]. A nuisance fire alarm is detected in a non-fire situation. Nuisance fire alarms have been reported as the main cause of smoke detectors being inferior. They lower the reliability of smoke detectors at the forefront of evacuations due to economic issues, waste of firefighting personnel effort, and lack of safety awareness [3,4]. In South Korea, the economic loss per unwanted fire alarm is KRW 474,892, with considerable economic and material losses occurring annually [5].

### 1.2. Research Trends and Purpose

Research on unwanted fire alarms by smoke detectors has been continuously conducted. Choi's study on unwanted fire alarms caused by cooking nuisances was conducted through full-scale experiments based on UL 268 [6]. The Consumer Product Safety Commission [7] installed photoelectric smoke, dual-sensor, and ionization detectors at 5-, 10-, and 20-foot intervals and reported that the sensors could be affected by various cooking methods, food types, and residential types. NIST [8] also investigated the characteristics of smoke detectors via experiments on unwanted fire alarms during the cooking of ingredients such as beef and bacon using various cooking methods. Zevotek [9] performed experiments on fire and UWFAs using ingredients such as fish and cooking oil. The Fire Protection Association (FPA) [10] performed experiments on UWFAs caused by water vapor and

toasters. In addition, in a recent study conducted by NASA, experiments using various sensors were conducted for early detection in spacecraft [11], and NIST also developed an algorithm using sensors such as CO for ignition prevention [12].

Based on the identified research trend, it has been recognized that various experiments have been conducted to address unwanted fire alarms. However, it is determined that there is a need for research and verification specifically focused on cooking nuisance false alarms that are increasingly observed in studio-type apartments [13,14]. Therefore, this study is the first to select and utilize indoor air quality sensors that can secure smoke detectors in order to reduce cooking fire alarms generated in a studio-type apartment.

## 2. Approach

### 2.1. Statistics of Unwanted Fire Alarms

Statistics related to unwanted fire alarms in the Republic of Korea, the United States (U.S.), and the United Kingdom (U.K.) between the years of 2018 and 2020 were examined. No statistics occur on non-fire reports in the Republic of Korea, and only the number of dispatches by mistake due to non-fire reports during daily safety dispatches is counted. To supplement this, the National Fire Agency collected statistics by classifying the causes and locations of non-fire alarms from March to May 2018. According to the data of the National Fire Agency [15], it was confirmed that the dispatches caused by unwanted fire alarms represent 6.81% of daily safety dispatches each year. In addition, the number of fire dispatches is decreasing, whereas the number of dispatches caused by unwanted fire alarms is increasing annually [16]. To supplement this information, the National Fire Agency (NFA) of Korea collected statistics by classifying the causes and locations of unwanted fire alarms over a short period of time. According to the 2019 NFA data [15], unwanted fire alarms were most commonly caused by artificial factors, including mischief and mistakes (16.8%) and cooking (8.8%). Unwanted fire alarms caused by cooking occur more frequently from smoke detectors than from heat detectors.

In the U.S., both fire and unwanted fire alarm statistics are collected nationwide via the National Fire Incident Report System (NFIRS) [17]. The number of unwanted fire alarm cases was more than 2,000,000 each year during 2018–2020. The number of unwanted fire alarms was 1.5 times larger than the number of fires each year. The main cause of unwanted fire alarms was unintentional calls (unintentional operation of systems, etc., in non-fire situations), which accounted for more than 50% of cases (1,143,287) from 2018 to 2020.

In the UK, unwanted fire alarm statistics are managed in an integrated manner [18]. The number of unwanted fire alarms during 2018–2020 represented more than 30% of the annual number of fires. The most common cause of unwanted fire alarms was the device (149,495 cases, 66.05%), followed by human action (67,142 cases, 44.91%). As observed through statistics, smoke detectors and human action are major causes of unwanted fire alarms. In particular, as per the statistics in the UK, more than 50% of the UWFAs caused by people were identified as cooking nuisance alarms.

Therefore, UWFAs occur most often from smoke detectors, and most of them are caused by cooking. Accordingly, the performance test method and installation location of the smoke detector are to be confirmed.

### 2.2. Smoke Performance Test

In order to prepare for the UWFAs of smoke detectors, various tests and standards occur in the Republic of Korea, the United States, and the United Kingdom. Among them, the standard for the UWFA test as a full-scale test is UL 268. In UL 268, fire tests and UWFA tests were conducted, which included a cooking nuisance smoke alarm test. The experiment was conducted while adjusting the length according to the type of sample in the test room (W6.7xL11xH3). The sample types used were paper, wood, inflammable fuel, and polyurethane foam, which are generally used in fire tests. An experiment using hamburger patties was conducted as an experiment related to UWFAs. In the cooking nuisance smoke alarm test, a non-fire alarm situation was created by using an oven with

hamburger patties. The detector installed at a distance of 3 m away from the sample must not operate in the test environment and must meet the corresponding profile conditions (CO, ODM: optical density meter, etc.) and method [19].

*2.3. Smoke Detector Installaion Positions*

In most countries, smoke detectors are installed in corridors, stairs, and living rooms/spaces. However, smoke detector installation positions may differ slightly by country. In Korea, smoke detectors are installed in the living room, which is used for sleeping, lodging, hospitalization, and other purposes, for specific firefighting targets that correspond to apartments, studio apartments, accommodations, welfare facilities, and training facilities [20]. In addition, places where installation cannot be performed (such as kitchens), places where similar smoke stays (such as bathrooms), and places where a large amount of smoke is generated are regulated, preventing UWFAs caused by cooking by-products. In the U.S., smoke detectors apply performance-based design in some cases, or NFPA 72 requires a distance of 30 ft. (9.1 m) $\pm$ 5% [18 in. (460 mm)] in residential facilities, and at the same time, for non-fire alarms. The cause is presented in a table and guidelines for installation are presented [21]. In the UK, guidelines are divided into BS 5839-1 and BS 5839-6 according to residential and non-residential spaces and require the type of sensor and installation interval according to the characteristics of the space. In addition, Chapter 12 separately mentions the causes of UWFAs with reminders to install detectors with an awareness of UWFAs [22,23].

In Korea, studio-type housing, with an area of less than 60 m$^2$, was renamed as small housing [24–26]. In the U.S., according to the "Maximum Occupancy of Dwelling Units for Sleeping Purposes" in California, the state with the largest population, the minimum residential space must be 17.7 m$^2$ or larger in accordance with IBC Section 1208.4, and a separate closet and separate bathroom with a shower facility must be provided [27]. In addition, according to the case in the "Maximum Occupancy of Dwelling Units for Sleeping Purposes," a bedroom of at least 220 ft$^2$ (20.44 m$^2$) and a separate kitchen and bathroom are required [28]. In the UK, the minimum area according to the number of people is presented through guidelines, and a minimum area of 37 m$^2$ is suggested for one person [29,30].

Each country has different regulations for the minimum and maximum dimensions of a studio-type apartment as well as different naming conventions. However, an area of at least 37 m$^2$ is commonly equired in addition to a separate bedroom, living room, kitchen, and bathroom, as shown in Table 1.

**Table 1.** Standards for studio-type housing according to country.

| Country | Designation | Statute | Regulation |
|---|---|---|---|
| Republic of Korea | Studio-type Housing | Enforcement Decree of The Housing Act Minimum Housing Recommendation | 14 to 60 m$^2$ Bathrooms and kitchens shall be installed so that each household can reside in such housing independently from other households. |
| U.S. | Studio-type Apartment | IBC—Section 1208.4 | 17.7 m$^2$ |
| | | Maximum Occupancy of Dwelling Units for Sleeping Purposes | 20.44 m$^2$ |
| UK | Studio-Flat | London's planning guidance | 37 m$^2$ |

In this way, a separate kitchen is required in the studio-type apartment, and a smoke detector must be installed, which results in UWFAs.

The results of examining studio-type apartments in each country and the sensors, samples, and test methods selected for the experiments were utilized to design the testbed for studio-type apartments. The total area of the testbed, including two rooms and passageways, was 30.5 m$^2$. In addition, the sensors (CO and PM10.0) and smoke detectors were selected to

examine the smoke detector arrangement method and precautions taken during installation. They were installed at seven positions and their characteristics were investigated.

## 3. Method

### 3.1. General Details

As shown in Figure 1 of this study, we reviewed literature to select a sensor that can support a smoke detector, and based on the UL 268 full-scale UWFA smoke detector experiment, we selected a cooking scenario wherein a frying pan was used and conducted the experiment. Based on the experimental results, we developed an algorithm that could determine UWFAs using indoor air quality sensors, and confirmed its effectiveness through verification experiments.

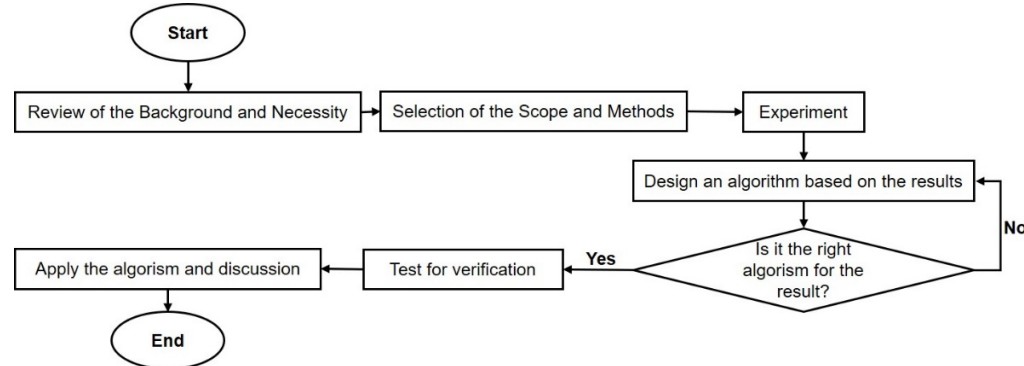

**Figure 1.** Flowchart for study processes.

### 3.2. Cooking Nuisance Smoke Alarm Test

The testbed had a length of 9.1 m, width of 6.7 m, and height of 3.0 m. Figure 2a,b show the drawings of the test room, through which the location of the heating element and measuring equipment can be confirmed. The experiment was performed based on the content selected based on theoretical considerations. The measuring instruments were installed on the ceiling at a distance of 3.05 m from the sample position. The spacing between the measuring instruments was 0.076 mm. Figure 2 shows technical drawings and a photograph of the testbed.

We selected frying as the method of cooking, which is the most common cooking method in various food cultures; it included putting a frying pan on a gas stove installed in the kitchen. In addition, the test room specifications and other test conditions were configured similarly to those used in the test of UL 268, which is a performance test of smoke detectors. Figure 3a shows the heating device and samples used during the test. A gas stove that used liquefied petroleum gas (LPG) was used as a heating device and installed on a kitchen counter. A coated frying pan with a diameter of 28 cm was used as a cooking tool. The total maximum gas consumption was less than 7800 kcal/h (9.1 kW), and the maximum gas consumption of one large burner was less than 3440 kcal/h (4 kW). In the experiment, 20 kg of LPG was used for domestic use, and its main component was propane ($C_3H_8$) with a purity of 99.9% or higher. The maximum LPG flow rate for the largest burner was less than 2.39 L/min, and the flow rate could be adjusted.

The samples used in the experiment were acquired in various forms of groceries such as meat (hamburger patty, bacon, pork belly, etc.), fish, and water. Among them, it was confirmed that smoke generation was sufficient in meat and fish; thin-sliced pork belly, which is the most consumed meat in Korea, and mackerel, which is a fish, were selected [31,32].

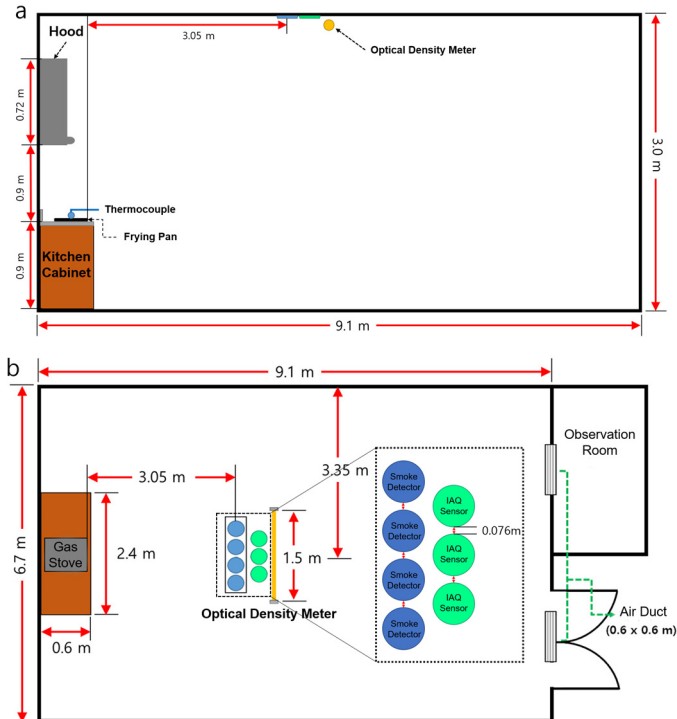

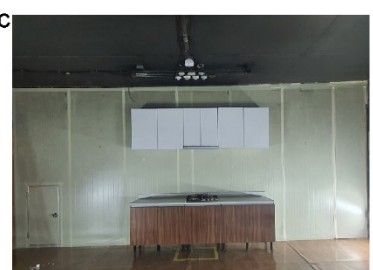

**Figure 2.** Testbed room drawings and photograph. Test room (**a**) side view, (**b**) floor plan, and (**c**) photograph.

In Figure 3b, the sample, thin-sliced pork belly, is shown. Regarding the samples, the thin-sliced pork belly was a product containing fat ($40 \pm 5\%$) and lean meat ($60 \pm 5\%$). It had a thickness of $2 \pm 0.5$ mm, a length of $120 \pm 0.5$ mm, a width of $40 \pm 0.5$ mm, and a total weight of $300 \pm 20$ g. In the case of mackerel, one fish with a length of $25 \pm 2$ cm and a total weight of $300 \pm 20$ g was used, which was split into two pieces. Each sample was frozen for 72 h in a refrigerator ($-20$ to $25\,^{\circ}$C) before use. For the mackerel, 30 g of cooking oil was added to prevent it from adhering to the pan during cooking.

Various sensors can determine fire and UWFAs. When smoke detectors operate, soot and smoke particles, as combustion products in the initial stage of fire (incipient fire), account for most of the elements, and the size of smoke particles is known to be approximately 0.01–10.0 μm [5,33]. As the by-products generated via cooking have a particle size similar to smoke in the early stage of a fire, they are not distinguishable from the standpoint of photoelectric smoke detectors, resulting in a false alarm [5]. As a result of reviewing previous studies, it was confirmed that various sensors such as CO, $CO_2$, NO, and $NO_2$ were used as sensors to improve photoelectric smoke detectors [6–10,19,34–39]. Among the CO and Particulate Matter (hereinafter referred to as PM) sensors that are judged to have some performance in UWFAs, PM10.0, which has the largest particle size, was selected. Figure 4 shows the sensors used in the tests. An optical density meter (ODM; Figure 4a) was installed to measure the obscuration-per-meter (OPM) profile of UL 268, whereas a K-type thermocouple (THCP; Figure 4b) was installed to measure the

temperature of the pan. Regarding the performance evaluation, two conventional smoke detectors (CSDs; marked as an area that is not an accurate location in the event of a fire) and two analog smoke detectors (ASDs; the exact location of the detector is checked), which are commonly used, were installed. For the indoor air quality sensor (IAQS; indoor air quality sensor), three units with the same performance were installed. As shown in Figure 4g, the IAQS contained CO and PM sensors inside the detector-type device. CO can be measured from 0 to 500 ppm, and up to 1000 $\mu g/m^3$ of PM can be measured based on ultrafine dust.

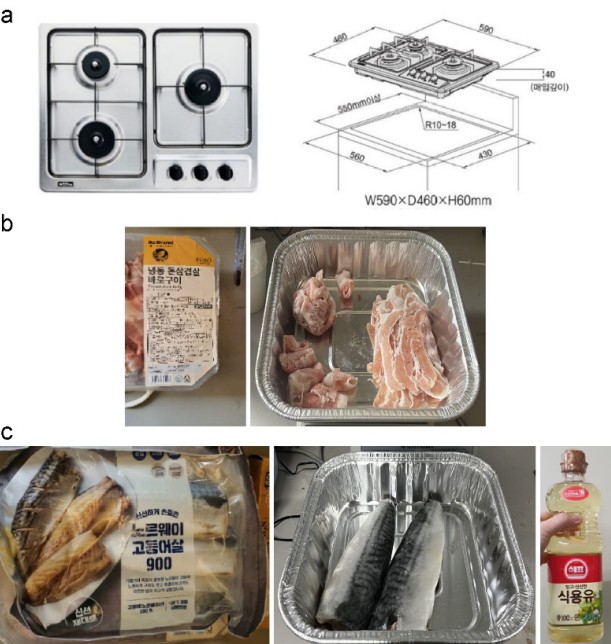

**Figure 3.** Heating device and samples: (**a**) heating device; (**b**) thin-sliced pork belly; (**c**) mackerel and cooking oil.

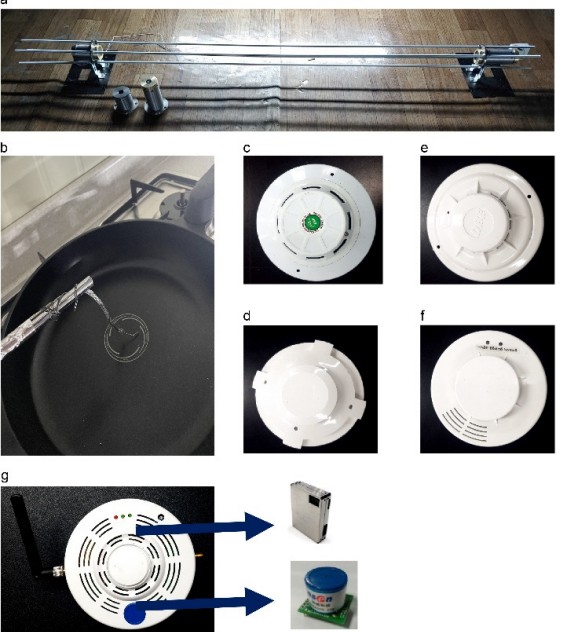

**Figure 4.** Measurement sensors: (**a**) optical density meter; (**b**) thermocouple; (**c**) manufacturer A (ASD); (**d**) manufacturer B (ASD); (**e**) manufacturer A (CSD) 15%; (**f**) manufacturer B (CSD) 15%; (**g**) indoor air quality sensor.

The environmental conditions followed the temperature, humidity, and wind speed conditions required for the UL 268 cooking nuisance smoke alarm test, and the experiment was repeated four times. The initial state for repeating the experiment was made into the same initial state for the ODM, CO, and PM through the ventilation hole. For the test, the prepared samples were placed in a frying pan, and the maximum firepower of the gas stove was maintained for 5 s. The samples were continuously heated without flipping over to minimize the test variables, and the test was conducted for 16 min. For the experiment using mackerel, cooking oil was applied to it, and the mackerel was placed with its skin facing down. Table 2 shows the test process and basic details, such as sample photographs and masses before and after the test.

**Table 2.** Basic experimental details.

| | Thin-Sliced Pork Belly | | | | Mackerel | | | |
|---|---|---|---|---|---|---|---|---|
| | **Case 1** | **Case 2** | **Case 3** | **Case 4** | **Case 1** | **Case 2** | **Case 3** | **Case 4** |
| The room temperature (°C) | 23.9 | 24.8 | 23.8 | 23.1 | 20.9 | 21.7 | 22.6 | 23.9 |
| Humidity (%) | 40.1 | 39.8 | 42.8 | 42.1 | 42.3 | 41.2 | 48.3 | 41.5 |
| Air current (m/s) | 0.0 | 0.0 | 0.0 | 0.0 | 0.0 | 0.0 | 0.0 | 0.0 |
| Before the experiment mass (g) | 300.2 | 290.9 | 309.6 | 302.4 | 309.0 * | 309.0 * | 318.2 * | 316.6 * |
| After the experiment mass (g) | 121.8 | 122.8 | 133.0 | 96.8 | 186.9 | 199.6 | 200.7 | 209.2 |
| Before the experiment | 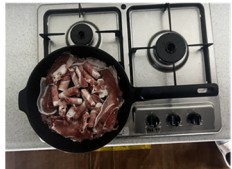 | | | | 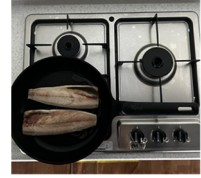 | | | |
| After the experiment | 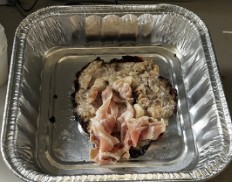 | | | | | | | |


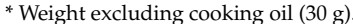

\* Weight excluding cooking oil (30 g).

### 3.3. Studio-Type Apartment Test

A testbed for a studio-type apartment testbed was designed by referring to previous studies on studio-type apartments in various countries.

The total area of the testbed was 30.5 m$^2$, and two rooms (each with width 4 × length 2.5 × height 2.5) and a corridor (width 7 × length 15 × height 2.5) were composed. In addition, in order to check the arrangement method of smoke detectors and precautions during installation, selected sensors (CO, PM10.0) and smoke detectors were installed in seven locations. Figure 5a,b show the drawing of the test rooms and the installation locations of the measurements. The same ODM as the cooking nuisance smoke alarm test was installed in two places, and the same THCP as the cooking nuisance smoke alarm test was installed to measure the fan temperature. The same detector as the cooking nuisance smoke alarm test was installed for each location, and the same IAQS was installed, one by one. Figure 5c is a photograph of each room. In addition, other environmental conditions were configured similarly to those in the cooking nuisance smoke alarm test. The test was terminated after the smoke detector was activated. Table 3 shows the basic details, such as the test process, before and after sample photos, and mass.

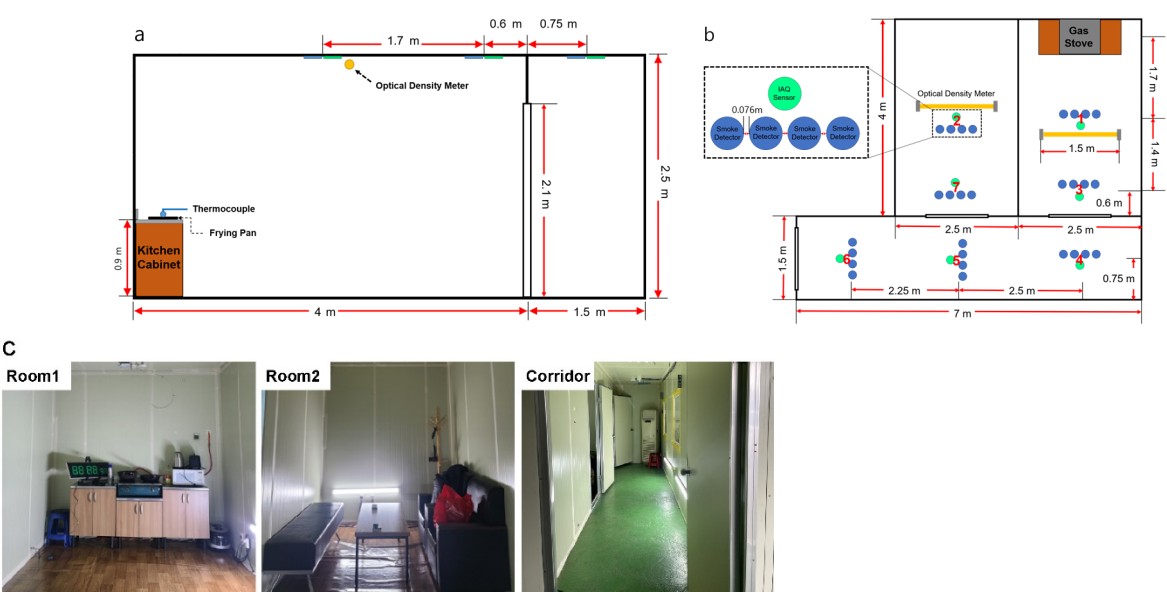

**Figure 5.** Studio-type apartment testbed (**a**) side view; (**b**) floor plan, and (**c**) photographs.

**Table 3.** Basic studio-type apartment test details.

|  | Thin-Sliced Pork Belly | Mackerel |
|---|---|---|
| The room temperature (°C) | 19.5 | 21.7 |
| Humidity (%) | 40.0 | 42.3 |
| Air current (m/s) | 0.0 | 0.0 |
| Mass before the verification experiment (g) | 302.3 | 309.9 |
| Mass after the verification experiment (g) | 122.8 | 174.6 |
| Before the verification experiment |  |  |
| After the verification experiment |  |  |

## 4. Results and Discussion

### 4.1. Cooking Nuisance Smoke Alarm Test

Data were analyzed using the average and standard deviation of the experimental measurements, which were repeated four times for each test. The results were analyzed in the following order: OPM tendency, detector response time, and IAQS (measurement range).

#### 4.1.1. Uniformity of Using Optical Density Meter (ODM) and Build-Up Time

The OPM tendency was examined using the ODM to identify the consistency of the experiment. Figure 6 shows the ODM measurement results for each experiment. It can be seen that the four experiments were performed by following similar patterns. In the experimental results for thin-sliced pork belly (Figure 6a), the average rate of OPM change

per minute was found to be a quadratic function, as follows: OPM = $0.17t^2 - 1.03t$, with the coefficient of determination ($R^2$) of 0.97. Also, in the experimental results for mackerel (Figure 6b), the average rate of OPM change per minute was found to be a quadratic function, as follows: OPM = $0.08t^2 - 0.19t$, with $R^2$ = 0.99. The average value range was found to be ±1.53 (thin-sliced pork belly) and ±0.88 (mackerel) from the maximum and minimum values at each time point in the entire test. This shows that the experiment proceeded consistently.

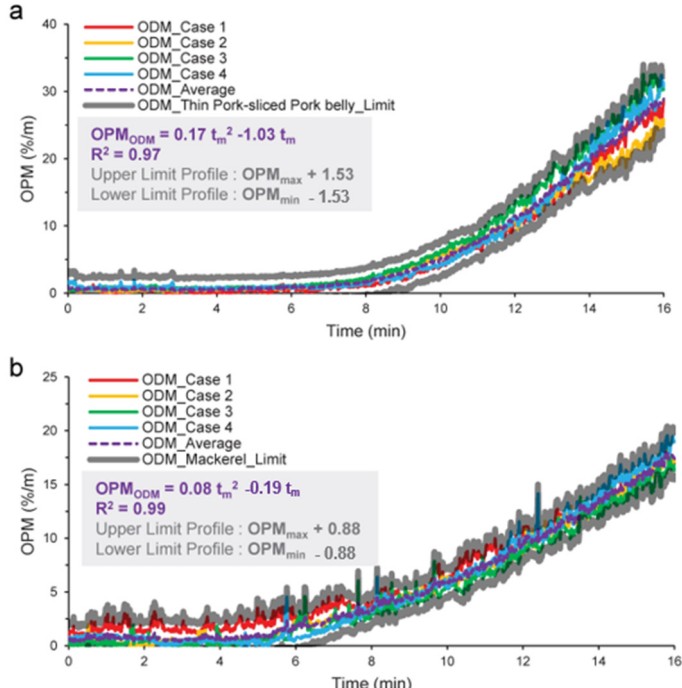

**Figure 6.** OPM with time measured by ODM for the (**a**) thin-sliced pork belly and (**b**) mackerel test.

Based on these results, we examined the specificity. In the experimental average change rate graphs, the time point when smoke began to form (hereafter, build-up) tended to be earlier in the mackerel experiment than in the thin-sliced pork belly experiment. The average temperature change on the surface of the pan used in each experiment is shown in Figure 7. In each experiment, the time to reach the boiling temperature of water (100 °C) was found to be 6.10 min for thin-sliced pork belly and 4.07 min for mackerel. Thus, it appears that build-up began after all the moisture contained on the surface and inside the sample evaporates. It was determined that the melting and evaporation processes occurred faster in the frozen state for mackerel because mackerel has a relatively higher moisture content than thin-sliced pork belly. That is, thermal diffusion becomes easier as the moisture content increases during heating at the same temperature and it becomes easier as the heating temperature increases at the same moisture content [40].

In the graphs, the smoke generation rate can be verified after the build-up point. It can be seen that thin pork had a higher change rate than mackerel. This is because smoke was generated as the sample surface made of fat and protein was subjected to the Maillard reaction after the evaporation of moisture. In turn, this occurs because thin pork, which has a larger surface area in contact with the frying pan, was more favorable in terms of heat transfer efficiency than mackerel, which has a smaller surface area in contact with the frying pan and a higher sample thickness with the same moisture content [40,41].

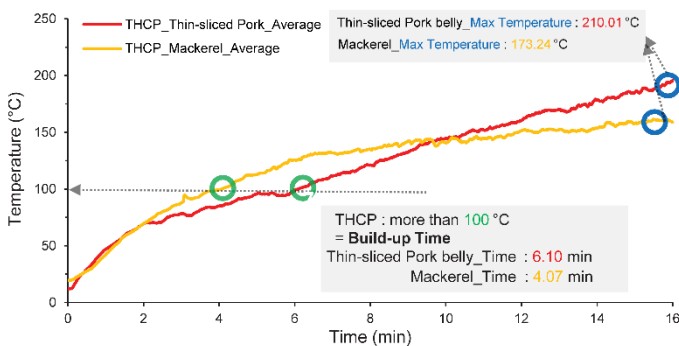

**Figure 7.** Pan temperature change by test.

### 4.1.2. The Activation Time of the Smoke Detector

The ODM and ASD were compared based on the operation of the CSD and 5%/m of the installed detector (ASD) (minimum standard for fire alarm). As shown in Figure 8, the ODM and ASD showed comparable trends in each test. The trends were similar, although the ODM and ASD measurements and change rates were slightly different. In the experimental results for thin-sliced pork belly shown in Figure 8a, a time difference of 4 min occurred between the ODM and ASD in the build-up period. The time difference was 3 min when the OPM was approximately 5%/m and 2 min when it was 10%/m or higher, showing that the time taken for the ODM to increase by 5%/m was reduced by 0.5 min due to smoke. A similar pattern was observed in the experimental results for the mackerel, as shown in Figure 7b. There was a time difference of 5 min between the ODM and ASD groups during build-up. The time difference was 4.5 min when the ODM was approximately 5%/m and 4 min when it was 10%/m or higher, showing that the time taken for the ODM to increase by 5%/m was reduced by 0.5 min due to smoke. This appears to be the pattern caused by the difference in thickness and surface area subjected to heat transfer between the samples discussed earlier.

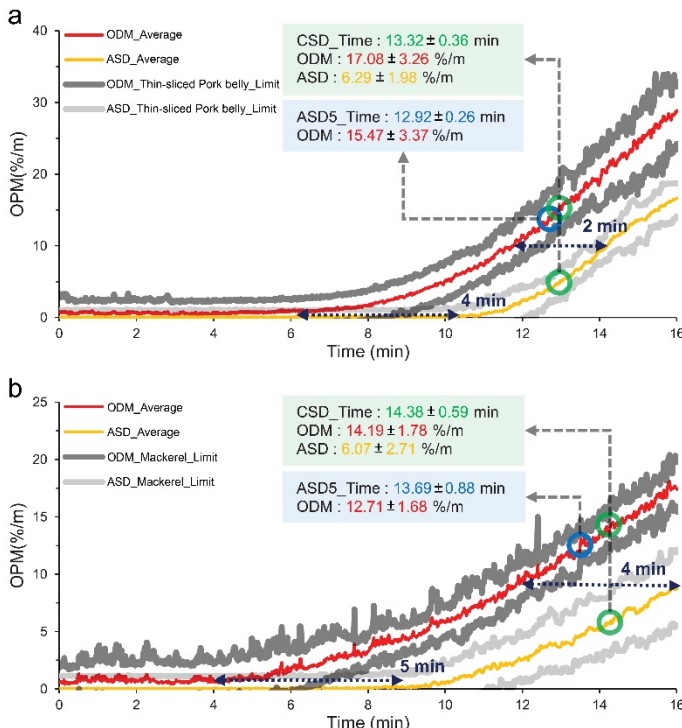

**Figure 8.** Comparison between ODM and ASD according to CSD and ASD activation time for the (**a**) thin-sliced pork belly and (**b**) mackerel tests.

In addition, as shown in Table 4, the CSD activation point and the time when the ASD reached 5%/m OPM were identified.

**Table 4.** Fire smoke detector operation time during the performance test.

| Specimen | Detection Time (min) | Manufacturer A | | | | Manufacturer B | | | | Avg. | STDEV.P |
|---|---|---|---|---|---|---|---|---|---|---|---|
| | | Case 1 | Case 2 | Case 3 | Case 4 | Case 1 | Case 2 | Case 3 | Case 4 | | |
| Thin-sliced Pork Belly | CSD | 13.88 | 13.40 | 12.70 | 13.80 | 13.17 | 13.20 | 13.03 | 13.38 | 13.32 | 0.36 |
| | ASD (5%/m) | 13.03 | 13.08 | 12.60 | 13.33 | 12.68 | 12.58 | 12.90 | 13.13 | 12.92 | 0.26 |
| Mackerel | CSD | 15.08 | 14.40 | 15.32 | 13.70 | 14.35 | 14.40 | 14.37 | 13.38 | 14.38 | 0.59 |
| | ASD (5%/m) | 14.90 | 13.37 | 14.52 | 12.37 | 14.35 | 13.38 | 14.17 | 12.47 | 13.69 | 0.88 |

Table 4 shows the activation time of the CSD and the time when the ASD reached 5%/m. In the thin-sliced pork belly test, the CSD activation point was found to be $13.32 \pm 0.36$ min and the time when the ASD reached 5%/m was $12.92 \pm 0.26$ min. In the mackerel test, the CSD activation point was $14.38 \pm 0.59$ min and the time point when the ASD reached 5%/m was $13.69 \pm 0.88$ min. The results tended to be delayed in the mackerel test compared to those in the thin-sliced pork belly test. At the CSD activation point, however, it was found that the OPM of thin-sliced pork belly was $17.08 \pm 3.26$%/m while that of mackerel was $14.19 \pm 1.78$%/m, and that the ASD of thin-sliced pork belly was $6.29 \pm 1.98$%/m while that of mackerel was $6.07 \pm 2.71$%/m. At the time when the ASD reached 5%/m, there was also a difference of $15.47 \pm 3.37$%/m for thin-sliced pork belly and $12.71 \pm 1.68$%/m for mackerel.

The delay of the CSD activation in the mackerel experiment compared to the thin-sliced pork belly experiment while having a larger ODM value is due to the characteristics of photo-scattering-type smoke detectors, which have more chance of activation with smaller smoke particles [42]. It was determined that less time was required for smoke particles from thin-sliced pork belly to penetrate the sensor compared to those from mackerel because of the relatively small size, high smoke generation rate, and larger amount of smoke; however, photo-scattering in the sensor occurred later compared to the mackerel due to the smaller size of the smoke particles.

### 4.1.3. Indoor Air Quality Sensor

At the CSD operating point and the time when the ASD reached 5%/m, the CO and PM10.0 values measured by the IAQS were examined. Figure 9 shows the CO measurement results. Figure 9a,b show the results for thin-sliced pork belly and mackerel, respectively. The results were found to be $2.52 \pm 1.04$ and $2.98 \pm 1.75$ ppm for thin-sliced pork belly and mackerel, respectively, at the CSD operating point and $2.63 \pm 0.93$ and $2.91 \pm 1.73$ ppm for thin-sliced pork belly and mackerel, respectively, at the time when the ASD reached 5%/m OPM. The CO range for thin-sliced pork belly was narrower than that of mackerel.

Figure 10 shows the PM10.0 measurement results. Figure 10a,b show the results for thin-sliced pork belly and mackerel, respectively. The results were found to be $5834 \pm 1418$ and $5668 \pm 1301$ $\mu g/m^3$ for thin-sliced pork belly and mackerel, respectively, at the CSD operating point and $5390 \pm 1384$ and $5333 \pm 1221$ $\mu g/m^3$ for thin-sliced pork belly and mackerel, respectively, at the time when the ASD reached 5%/m OPM. The PM10.0 measurement range for thin pork was also found to be different from that of mackerel.

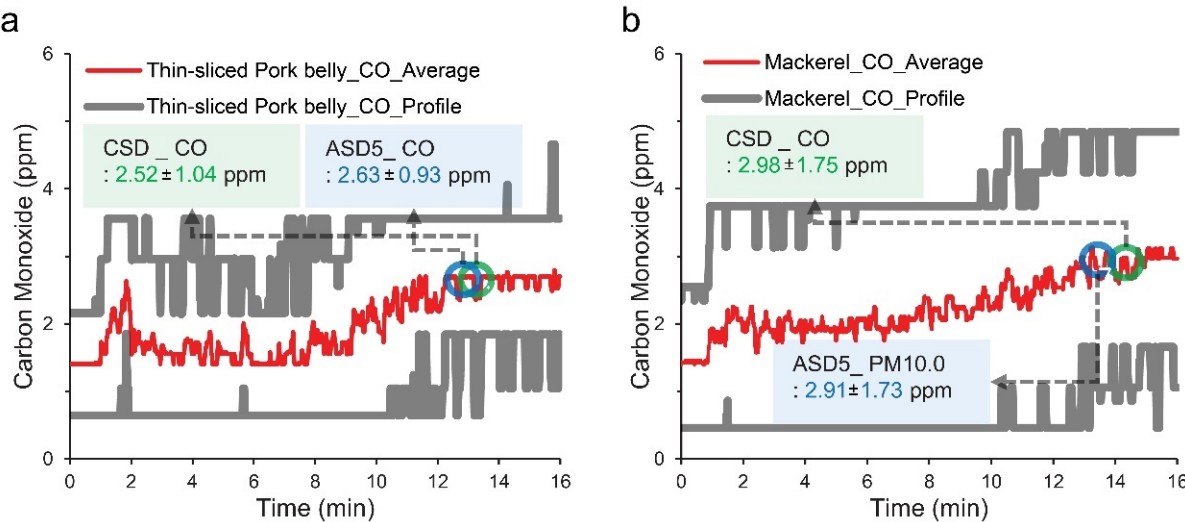

**Figure 9.** CO measurement results for the (**a**) thin-sliced pork belly and (**b**) mackerel tests.

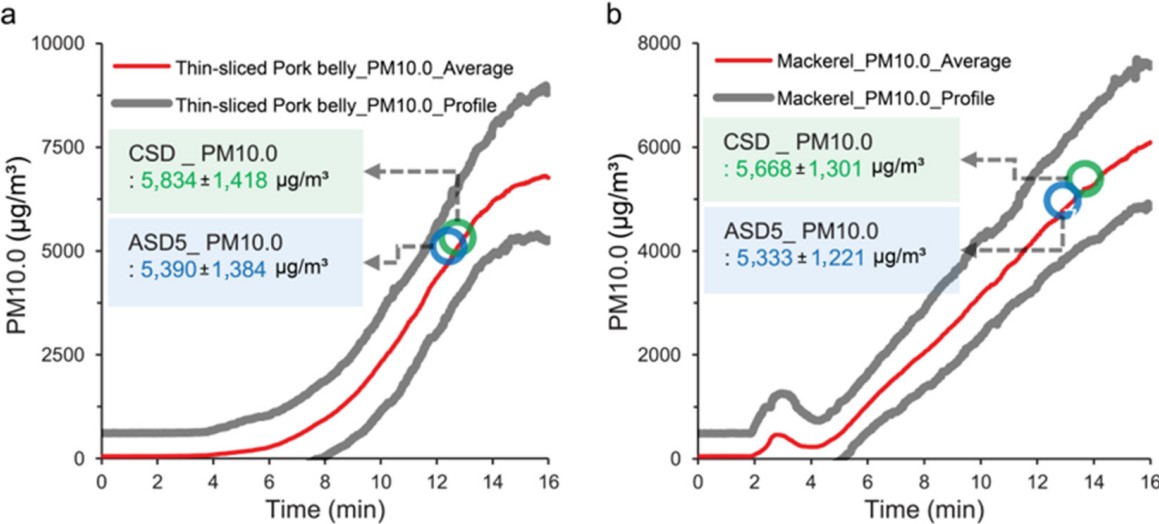

**Figure 10.** PM10.0 measurement results for the (**a**) thin-sliced pork belly and (**b**) mackerel tests.

The CO and PM10.0 measurement results showed that the values at the CSD operating point and at the time when the ASD reached 5%/m were consistent, despite slight differences in overall change and measurement time according to sample.

### 4.1.4. Proposed Algorithm for Unwanted Fire Alarms

To propose an algorithm that utilizes CO and PM10.0 as indicators for determining unwanted fire alarms, the CO and PM10.0 measured through the IAQS were compared with the OPM values measured through the ODM and ASD, as shown in Figures 11 and 12. This is to compare values at the same time relative to the activation of the detector. Figure 11a compares the OPM of ODM with that of CO, whereas Figure 11b compares the OPM of the ASD with the CO. It can be seen that the CO measurements according to the OPM change generally show similar patterns regardless of the sample. Based on this, a criterion for determining unwanted fire alarms based on CO concentration was proposed by examining the overall distribution of the values, as shown in Figure 11c. A CO value of 5 ppm or less was proposed as the criterion for ASD 5%/m and ODM 15%/m. Figure 12a compares the OPM of ODM with PM10.0, whereas Figure 12b compares the OPM of ASD with PM10.0. It can be seen that the PM10.0 measurements according to the OPM change also generally exhibit similar patterns regardless of the sample. Based on this, a criterion for determining

unwanted fire alarms based on PM10.0 concentration was proposed by examining the overall distribution of the values, as shown in Figure 12c. A PM10.0 value of 7000 μg/m$^3$ or less was proposed as the criterion for ASD 5%/m and ODM 15%/m. In other words, if the CSD operates when the CO value is 5 ppm or less and the PM10.0 value is 7000 μg/m$^3$ or less, it can be determined to be an unwanted fire alarm. Figure 13 shows the proposed algorithm that represents the relationship among OPM, CO, and PM10.0. Moreover, the threshold values are safe levels under fire conditions [6].

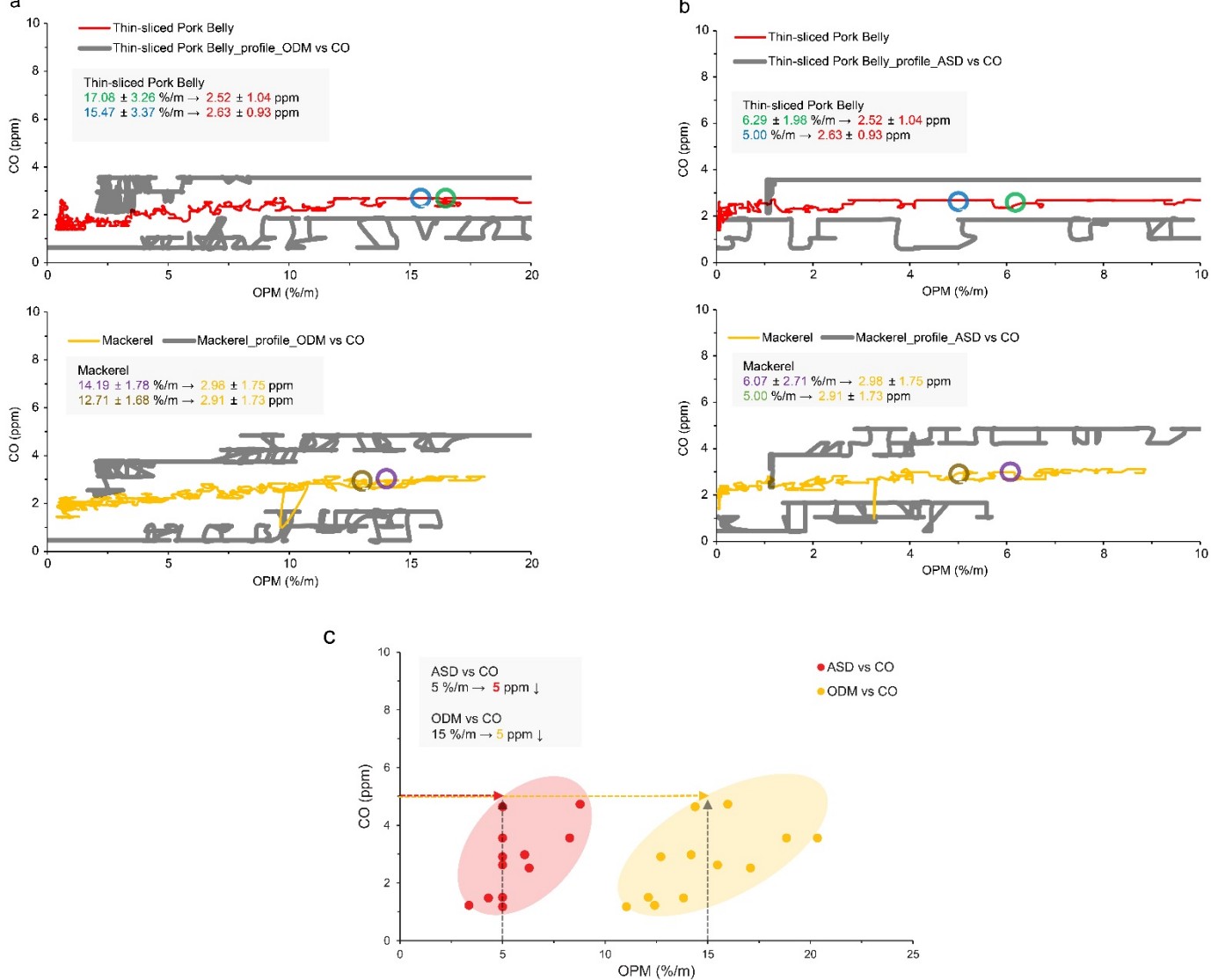

**Figure 11.** Comparison of the OPM and CO measurements: (**a**) ODM and CO; (**b**) ASD and CO; (**c**) OPM and CO combined results.

### 4.2. Studio-Type Apartment Test

A verification test was conducted to determine whether the measurements and ranges proposed in the performance test were appropriate.

#### 4.2.1. Consistency of Testing through ODM and Build-Up Time

Table 5 presents the ODM, ASD, CO, and PM10.0 values measured at each position through the verification test when the CSD was activated and the ASD reached 5%/m. Figure 14 compares the THCP between the performance and verification tests. First, during the verification test, the build-up time of thin-sliced pork belly was found to be 5.20 min, which was 0.9 min (54 s) earlier compared to the performance test. In the case of mackerel,

it was found to be 3.40 min, which was 0.67 min (40.2 s) earlier. Unlike the build-up time, the time to reach the maximum THPC value was delayed by 6.82 and 13.84 min for thin-sliced pork belly and mackerel, respectively, in the verification test compared to the performance test. Mackerel also reached the build-up time faster than thin-sliced pork belly in the verification test, similar to that in the performance test, and there was no significant difference (less than 1 min) between the samples.

**Figure 12.** Comparison of the OPM and PM10.0 measurements: (**a**) ODM and PM10.0; (**b**) ASD and PM10.0; (**c**) OPM and PM10.0 combined results.

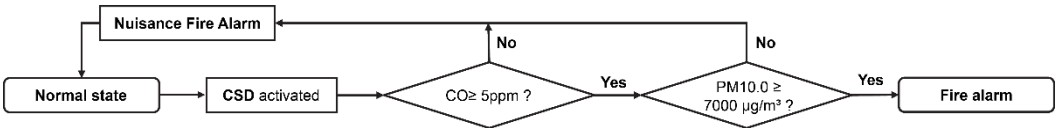

**Figure 13.** Proposed algorithm representing the relationship among OPM, CO, and PM10.0.

**Table 5.** Measurement values according to CSD operation and ASD 5%/m reached during the studio-type apartment test.

| Content | | P1 | P2 | P3 | P4 | P5 | P6 | P7 |
|---|---|---|---|---|---|---|---|---|
| **CSD** | ODM (%/m) Thin-sliced Pork Belly | 17.40 | 15.65 | - | - | - | - | - |
| | ODM (%/m) Mackerel | 7.73 | 10.65 | - | - | - | - | - |
| | ASD (%/m) Thin-sliced Pork Belly | 9.4 | 7.5 | 7.1 | 7.2 | 8.0 | 7.6 | 9.7 |
| | ASD (%/m) Mackerel | 14.3 | 5.4 | 10.1 | 9.0 | 7.6 | 6.6 | 10.1 |
| | CO (ppm) Thin-sliced Pork Belly | 4.1 | 4.3 | 3.1 | 4.1 | 3.9 | 3.7 | 3.1 |
| | CO (ppm) Mackerel | 4.1 | 2.6 | 2.6 | 1.6 | 2.8 | 1.0 | 2.4 |
| | PM10.0 ($\mu g/m^3$) Thin-sliced Pork Belly | 6956 | 5461 | 6923 | 6982 | 7169 | 6661 | 6922 |
| | PM10.0 ($\mu g/m^3$) Mackerel | 4891 | 4368 | 4432 | 4366 | 4530 | 4417 | 5109 |
| **ASD_5%** | ODM (%/m) Thin-sliced Pork Belly | 11.55 | 8.62 | - | - | - | - | - |
| | ODM (%/m) Mackerel | 6.62 | 9.79 | - | - | - | - | - |
| | CO (ppm) Thin-sliced Pork Belly | 3.0 | 4.3 | 3.1 | 4.1 | 3.9 | 3.7 | 3.1 |
| | CO (ppm) Mackerel | 4.1 | 2.6 | 1.4 | 1.6 | 1.8 | 0.9 | 2.4 |
| | PM10.0 ($\mu g/m^3$) Thin-sliced Pork Belly | 5900 | 5595 | 6032 | 7038 | 6030 | 6015 | 5872 |
| | PM10.0 ($\mu g/m^3$) Mackerel | 2533 | 4278 | 2789 | 2646 | 2947 | 3612 | 3394 |

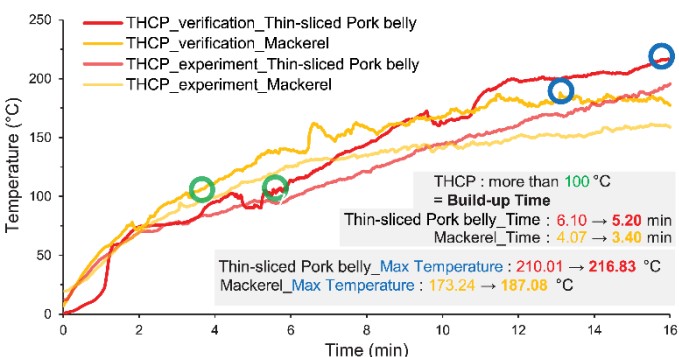

**Figure 14.** THCP verification test results.

4.2.2. Apply the Algorithm and Discussion

In the case of CO, the measurements were less than 5 ppm, a value proposed based on ASD 5%/m and ODM 15%/m, as shown in Figure 15a. As shown in Figure 15b, the ASD activated when the ODM measurement was 15%/m or less in most cases and it also activated when the CO measurement was less than 5 ppm. For the ASD that activated above 15%/m, the CO measurement was also found to be less than 5 ppm.

Third, it was confirmed that the proposed 7000 $\mu g/m^3$ or less for PM10.0 was appropriate. As shown in Figure 16a, the PM10.0 measurement was 7000 $\mu g/m^3$ or less within the upper and lower limits of ASD 5%/m and ODM 15%/m. It was also confirmed that the PM10.0 measurement was mostly 7000 $\mu g/m^3$ or less even when the ODM exceeded 15%/m, as shown in Figure 16b. In the case of thin-sliced pork belly, however, some measurements exceeded 7000 $\mu g/m^3$, as shown in Table 5, but they were within 10%.

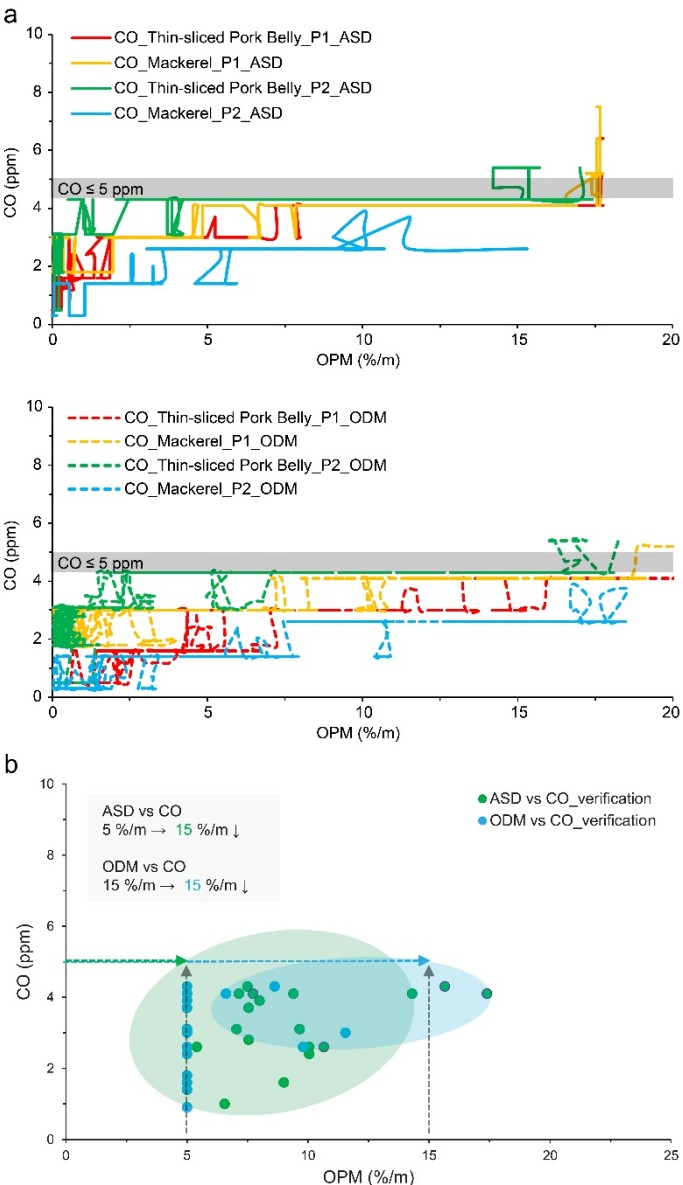

**Figure 15.** CO verification test results: (**a**) comparison with the performance test results; (**b**) verification test results.

### 4.3. Limitations and Complements of This Study

Although this study included several experiments and verification tests, it had limitations such as sample limitations and lack of diversity in test methods. Although the tests were conducted using meat (thin-sliced pork belly) and fish (mackerel), we could not confirm the same results using other samples. However, through the verification test (studio-type apartment), it was confirmed that the results did not vary greatly depending on the sample type. The test method (heating device) was also similar to the full-scale test method used in previous studies [6–10,19,37–42]. The test was conducted using frying, which is the most commonly used cooking method in households and involves emission of the highest amount of smoke particles, as compared to that in other methods such as boiling, steaming, poaching, and blanching. In future experiments, we wish to compare the effects of deep-frying and shallow-frying methods to confirm that a deep-frying method may more likely cause an unwanted fire alarm than a shallow-frying method.

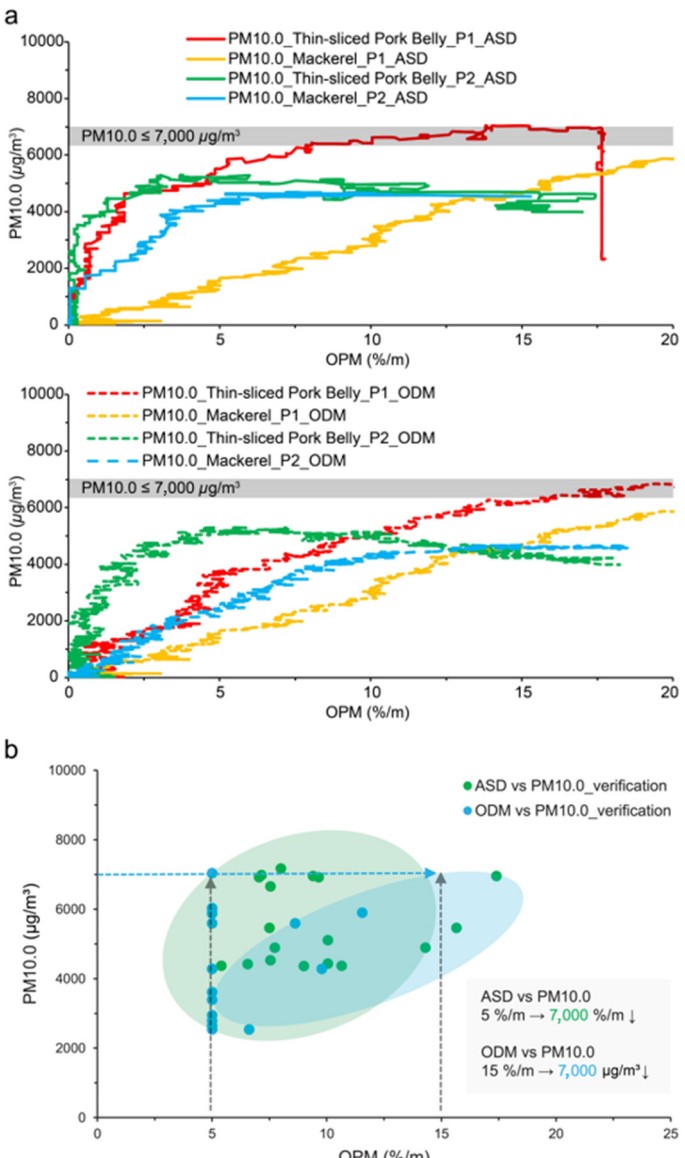

**Figure 16.** PM10.0 verification test results: (**a**) comparison with the performance test results; (**b**) verification test results.

## 5. Conclusions

An algorithm was proposed and verified using indoor air quality sensors to improve unwanted fire alarms caused by cooking by-products in studio-type apartments, which are blind spots for smoke detectors. The measurements and ranges of the proposed Indoor Air Quality Sensor (IAQS) were verified based on experimental results and considerations. The following conclusions were drawn.

Firstly, there was no significant difference in build-up time and smoke generation rate, based on the obscuration per meter (OPM) values at the operating point of the conventional smoke detector (CSD) and the time when 5%/m OPM was reached. This was confirmed across different samples, as the differences fell within the margin of error. It was observed that the probability of unwanted fire alarms generated by smoke detectors increased as particle size increased and particle density decreased. Fish, in particular, exhibited larger particles and lower particle density compared to meat.

Secondly, a criterion was proposed where the CO value should be 5 ppm or less after CSD activation to determine unwanted fire alarms. Performance and verification tests confirmed that the measured CO levels were 5 ppm or less when unwanted fire alarms

were generated for both meat and fish. It was specifically noted that the CO measurements were 5 ppm or less in all cases where the analog smoke detector (ASD) indicated 5%/m (the non-operational testing criterion) and the optical density meter (ODM) indicated 15%/m (the operational criterion).

Thirdly, a PM10.0 value of 7000 μg/m$^3$ or less after CSD operation was proposed as a criterion to determine unwanted fire alarms. The verification test also confirmed that the PM10.0 measurements were 7000 μg/m$^3$ or less, or within 10%, in all cases where the ASD indicated 5%/m and the ODM indicated 15%/m.

**Author Contributions:** Conceptualization: D.-m.C.; methodology: D.-m.C. and E.-h.H.; validation: D.-m.C., E.-h.H. and H.-b.C.; formal analysis: H.-b.C.; investigation: H.-b.C.; data curation: H.-b.C.; writing—original draft preparation: H.-b.C.; writing—review and editing: D.-m.C. and E.-h.H.; visualization: H.-b.C.; supervision: D.-m.C.; project administration: D.-m.C. All authors have read and agreed to the published version of the manuscript.

**Funding:** This work was supported by the Disaster Safety Platform Technology Development Program of the National Research Foundation of Korea funded by the Ministry of Science and ICT of the Republic of Korea (No. NRF-2019M3D7A1095926).

**Institutional Review Board Statement:** Not applicable.

**Informed Consent Statement:** Not applicable.

**Data Availability Statement:** All data generated or analyzed during this study are included in this published article.

**Acknowledgments:** Not applicable.

**Conflicts of Interest:** The authors declare no conflict of interest. The funders had no role in the design of the study; in the collection, analyses, or interpretation of data; in the writing of the manuscript; or in the decision to publish the results.

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
