# Peer review of "Indoor Air Quality Sensor Utilization for Unwanted Fire Alarm Improvement in Studio-Type Apartments"

_fire, doi:10.3390/fire6070255_

Round 1
Reviewer 1 Report

The English language throughout the paper needs improvement. I recommend the authors carefully revise the manuscript to ensure correct grammar, syntax, and clarity of expression. The text in some sections appears to be repetitive and lacks a cohesive flow, please try to be more concise.
Author Response
Thank you for your review.
We totally agree with you. And we have revised the manuscript.
Since multiple files cannot be transferred from the system, we will send you the answer to the review as follows. Also, please note that the attached file is a revised manuscript. The cover-letter and picture files are provided by Ms. I mailed it to Kristy Weid. Please confirm. Thank you very much.
Response to Reviewer 1 Comments
Thank you for seeking assistance in improving the quality of our work. In response to reviewer’s comments, we have made amendments to the following documents.
Title Changes: Indoor Air Quality Sensor Utilization for Unwanted Fire Alarm Improvement in Studio-type Apartment
→ The title modified to better represent the purpose of the paper.
Point 1: The English language throughout the paper needs improvement. I recommend the authors carefully revise the manuscript to ensure correct grammar, syntax, and clarity of expression..
Response 1: we have corrected grammar and syntax issues throughout the document. We apologize again for our mistake.
Point 2: Discussion: This section requires improvement as it appears repetitive and lacks a cohesive flow. I suggest the authors revise this section to present the findings in a logical and concise manner, avoiding unnecessary repetition.
Response 2: In the text, unnecessary repetition has been avoided, and the overall structure of the paper has been revised.
Point 3: Approach for Verification: To strengthen the study's validity, I recommend the authors consider conducting experiments with various types of food, rather than solely focusing on pork belly and mackerel. By demonstrating the applicability of the proposed thresholds to different food types, the authors can provide stronger evidence for their conclusions. This would significantly enhance the robustness of the study and broaden its relevance.
Response 3: We fully understand and acknowledge your concerns. Prior to writing our work, we conducted experiments involving vegetables, seasoned meats, and bread. However, verifying the consistency of each experimental value proved challenging, and after observing variations between samples, it was determined that the planer pork belly and mackerel used in the study are representative. The aforementioned information has been omitted from the paper for the sake of brevity. However, if necessary, we provide comments or suggest adding relevant references to address this issue.
Point 4: · Theoretical considerations can be incorporated into the introduction to provide a comprehensive background and context for the research.
Response 4: The sentence has been modified to align with your comment
Point 5: The section titled "Research Scope and Methods" is unnecessary and can be merged into the "Research Purpose" section. The methods used should be described in a separate section titled "Methods."
The research scope and method were adjusted to align with the research objectives,
Response 3: The research scope and method were adjusted to align with the research objectives, and the corresponding section has been revised to "Method."
Point 6: The section labeled "Experiment" should be renamed "Methods" and should encompass all information related to the methodology used in testing and validation experiments.
Response 6: The experiment was transformed into a method, and the methodology, as well as the equipment used for testing and verification experiments, were all described.
Point 7: A separate section titled "Results and Discussion" should be created to present and discuss the results obtained from both the testing and validation experiments.
Response 7: A section dedicated to results and discussion was created.
Point 8: Implementing these structural changes will improve the clarity and organization of the paper, allowing readers to follow the authors' research more effectively.
Response 8: We apologize for the inaccurate wording. Thresholds have already been taken into account at reference 6.

Reviewer 2 Report
To address the issue of unwanted fire alarms, this study conducted research aimed at reducing unwanted fire alarms caused by cooking nuisances when smoke detectors with a proper performance are exposed to unsuitable environments, such as in studio-type apartments. This work is quite useful and interesting, the comments can be found below,
1. Why do you choose CO and PM10.0 values as the judgment criteria for unwanted fire alarms caused by cooking nuisances? More reasons and explanations should be given.
2. The novelty of your work should be clear and additionally highlighted, together with the objectives of your research, in the last paragraph of the Introduction.
3. The background of this study is lacked in Abstract and described not enough in Introduction.
4. What are the limitation and uncertainty in the experiments? What is the measurement error?
5. There are quite a few grammatical and editorial errors in the text.
6. Please explain the reason why CO in Fig. 10 shows chaotic with OPM, what's the uncertainty of these data?
Moderate editing of English language required
Author Response
Thank you for your review.
We totally agree with you. And we have revised the manuscript.
Since multiple files cannot be transferred from the system, we will send you the answer to the review as follows. Also, please note that the attached file is a revised manuscript. The cover-letter and picture files are provided by Ms. I mailed it to Kristy Weid. Please confirm. Thank you very much.
Response to Reviewer 2 Comments
Thank you for seeking assistance in improving the quality of our work. In response to reviewer’s comments, we have made amendments to the following documents.
Title Changes: Indoor Air Quality Sensor Utilization for Unwanted Fire Alarm Improvement in Studio-type Apartment
→ The title modified to better represent the purpose of the paper.
Point 1: Why do you choose CO and PM10.0 values as the judgment criteria for unwanted fire alarms caused by cooking nuisances? More reasons and explanations should be given.
Response 1: From the past to the present, CO and PM have been used as criteria for judging many unnecessary fire alarms and fires. Relevant information has been described and references have been added.
Point 2: The novelty of your work should be clear and additionally highlighted, together with the objectives of your research, in the last paragraph of the Introduction.
Response 2: Thanks for improving the quality of the text. Each graph was modified by dividing it into two parts.
Point 3: The background of this study is lacked in Abstract and described not enough in Introduction.
Response 3: The introduction and abstract have been revised. Thank you.
Point 4: What are the limitation and uncertainty in the experiments? What is the measurement error?
Response 4: Added to Section 4.3 as comment. We hope that these additions meet your satisfaction. If there is anything else you would like us to include, please let us know. Thank you.
Point 5: There are quite a few grammatical and editorial errors in the text.
Response 5: we have corrected grammar and syntax issues throughout the document. We apologize again for our mistake.
Point 6: Please explain the reason why CO in Fig. 10 shows chaotic with OPM, what's the uncertainty of these data?
Response 6: We have explained in Fig. 10 the reason why CO changes with OPM at Section 4.1.4. As the structure of the paper has been revised, please note that the numbering of figures has also changed.

Reviewer 3 Report
All figures are missing in the article. Therefore, the figures and the paragraphs in which they are referred to cannot be evaluated. Therefore, a large part of section 3 and 4 is not sufficiently evaluable.
Section 1.1: Line 24-26 - Imprecise and incorrect definition of the purpose and functionality of smoke detectors. Smoke detectors can’t detect fires in advance. They are not clairvoyants. Please orientate on common definitions see ISO 7240-1:2014-06, Fire detection and alarm systems - Part 1: General and definitions; DIN CEN/TS 54-14:2019-01/ DIN SPEC 14002:2019-01, Fire detection and fire alarm systems - Part 14: Guidelines for planning, design, installation, commissioning, use and maintenance, Festag, S., 04/2023, Minimization of Risk by the Controlled Replacement of Fire Detectors, Fire Technology
Line 43-44 - Please specify the period for the collection of the causes and locations of unwanted fire alarms
Line 55 – What is the definition of unintentional calls? Please provide a precise definition and the complete statistical data for this
Section 2.1
Line 94-104 - The listed literature on studies around the topic of unwanted alarms here is far too old, no longer up to date and have no added value at this point. Citing a study on photoelectric smoke alarms from 1968, which represents the state of the art already for a long time, is not up to date. There are enough current studies from the 21st century in the USA and Great Britain that already deal with this topic as you mentioned below.
Section 2.2
Line 180-182 - I think the reason why in many countries fire detectors should not be installed in kitchens may have less relation to the high fire risk of kitchens and more to the vulnerability to false (esp. deceptive) alarms.
Section 3
Line 245-246 – Definition and explanation of analog smoke detectors and convectional smoke detectors necessary
Line 251 - There needs to be a brief explanation what the UL 268 regulations, specifications and test setups are. Not every reader will be familiar with this. In addition, this is part of the general test description, even if it is based on a standard.
Line 270-276 - A more detailed explanation of how these equations were created is necessary
Line 302-320 - Inferences described too imprecisely. More detailed and understandable explanations absolutely necessary
Section 3.3
Line 367 – 386 - Difficult to read due to a lot of shortcuts. Should be written slightly more readable
Section 4.1
Line 396-408 – Explanation why the verification test took place should be moved from Section 4.2 to Section 4.1 and clarify in which points the verification tests differ from the performance test and why or if not why not.
Author Response
Thank you for your review.
We totally agree with you. And we have revised the manuscript.
Since multiple files cannot be transferred from the system, we will send you the answer to the review as follows. Also, please note that the attached file is a revised manuscript. The cover-letter and picture files are provided by Ms. I mailed it to Kristy Weid. Please confirm. Thank you very much.
Response to Reviewer 3 Comments
Thank you for seeking assistance in improving the quality of our work. In response to reviewer’s comments, we have made amendments to the following documents.
Title Changes: Indoor Air Quality Sensor Utilization for Unwanted Fire Alarm Improvement in Studio-type Apartment
→ The title modified to better represent the purpose of the paper.
Point 0: All figures are missing in the article. Therefore, the figures andthe paragraphs in which they are referred to cannot be evaluated. Therefore, a large part of section 3 and 4 is not sufficiently evaluable.
Response 0 : We connected Ms. Kristy Wei Assistant. And she said it was solve.
Point 1: Line 24-26 - Imprecise and incorrect definition of the purpose and functionality of smoke detectors. Smoke detectors can’t detect fires in advance. They are not clairvoyants. Please orientate on common definition
Response 1: The definition of a smoke detector has been added in the Introduction, and the intent of the thesis has been modified in detail.
Point 2: Line 43-44 - Please specify the period for the collection of the causes and locations of unwanted fire alarms
Response 2:. The collection period was specified. In addition, the position of statistics was modified to improve the quality of the text.
Point 3: Line 55 – What is the definition of unintentional calls? Please provide a precise definition and the complete statistical data for this
Response 3: We apologize for that. As requested, we have provided the definition. Andditionally, the Statistics we found had errors in the the Statistical analyses; so hence, we have modified this section that. Sorry about that
Point 4 : Line 94-104 - The listed literature on studies around the topic of unwanted alarms here is far too old, no longer up to date and have no added value at this point. Citing a study on photo electric smoke alarms from 1968, which represents the state of the art already for a long time, is not up to date. There are enough current studies from the 21st century in the USA and Great Britain that already deal with this topic as you mentioned below.
Response 4: As studies on the topic of unwanted alarms here were far too old, they were deleted. Also, Recent studies have been added.
Point 5: Line 180-182 - I think the reason why in many countries fire detectors should not be installed in kitchens may have less relation to the high fire risk of kitchens and more to the vulnerability to false (esp. deceptive) alarms.
Response 5: We agree with you. So, we have incorporated the contents into the thesis and removed or corrected any parts that could potentially lead to misunderstandings.
Point 6: Line 245-246 – Definition and explanation of analog smoke detectors and convectional smoke detectors necessary
Response 6: Explanations of CSD and ASD were added.
Point 7: Line 251 - There needs to be a brief explanation what the UL268 regulations, specifications and test setups are. Not every reader will be familiar with this. In addition, this is part of the general test description, even if it is based on a standard.
Response 7: For the convenience of readers, we have added an explanation about UL268 in section 2.2.
Point 8: Line 270-276 - A more detailed explanation of how these equations were created is necessary
Response 8: The formula was presented using a trend line to identify the trend over time of ODM.
Point 9: Line 302-320 - Inferences described too imprecisely. More detailed and understandable explanations absolutely necessary
Response 9: Redundant and unnecessary expressions were corrected throughout the manuscript. And the content has been clearly modified.
Point 10: Line 367 – 386 - Difficult to read due to a lot of shortcuts. Should be written slightly more readable
Response 10: Thank you for your comment. We modified it. The full name is included in the text for the purpose of readability.
Point 11: Line 396-408 – Explanation why the verification test took place should be moved from Section 4.2 to Section 4.1 and clarify in which points the verification tests differ from the performance-test and why or if not why not.
Response 11: The structure was slightly changed for the completeness of the text, and the part about the verification test (Studio-type) was also added to reflect the review opinions.

Reviewer 4 Report
This paper conducted research on how to reduce unwanted fire alarms caused by cooking nuisances for smoke detectors exposed to unsuitable environment. Based on the experimental results, CO (5 ppm) and PM10.0 (7,000 μg/m3) values were proposed as the judgment criteria for unwanted fire alarms caused by cooking nuisances. Results show that the probability of a smoke detector which generate unwanted fire alarms would increase as the particle size increases and their density decreases. For determining unwanted fire alarms, a CO value of 5 ppm or less after CSD operation was proposed as a criterion and a PM10.0 value of 7,000 μg/m3 or less after CSD operation was proposed as a criterion. The results of the manuscript are of great value for the follow-up research. The manuscript can be published with minor revision.
1) In section 3.1, the ventilation of testbed site should be described in detail because it will influence the IAQS and PM value distribution. Is the experimental site chosen by the author representative?
2) The curves in Figures 10a,10b, 11a,11b, 15a and 16a cannot be seen clearly, it is recommended to redraw them.
Smoke detectors are directly involved in evacuations and fire safety. They play an important role and have a significant influence on the reliability, accuracy, and effectiveness of firefighting activities. Owing to improper non-defective smoke detector installation, unwanted fire alarms occur worldwide. To address this issue, in this study, we conducted research aimed at reducing unwanted fire alarms caused by cooking nuisances when smoke detectors with a proper performance are exposed to unsuitable environments, such as in studio-type apartments. Experimental conditions and an indoor air quality sensor (IAQS) were selected by examining related literature and standards; then, performance and verification tests were conducted accordingly. Based on the experimental results, CO (5 ppm) and PM10.0 (7,000 μg/m3) values were proposed as the judgment criteria for unwanted fire alarms caused by cooking nuisances.
Literature and standards were reviewed as theoretical considerations, and an experiment was conducted to determine ways to utilize an IAQS to improve unwanted fire alarms via cooking nuisance sources. The measurements and ranges of the proposed
IAQS were verified based on experimental results and considerations. The following clusions were drawn.
First, as for the obscuration per meter (OPM) values at the operating point of the conventional smoke detector (CSD) and at the time when 5%/m OPM was reached, there was no significant difference in build-up time and smoke generation rate according to sample because the difference was within the margin of error. It was confirmed that the probability that a smoke detector will generate unwanted fire alarms increases as the particle size increases and their density decreases, and that fish has larger particles and lower particle density than meat.
Second, a CO value of 5 ppm or less after CSD operation was proposed as a criterion for determining unwanted fire alarms. Performance and verification tests confirmed that the measured CO was 5 ppm or less when unwanted fire alarms were generated for both meat and fish. In particular, it was confirmed that the measurement of CO was 5 ppm or less in all cases where the analog smoke detector (ASD) was 5%/m, which is the judgment criterion for nonoperational testing, and the optical density meter (ODM) was 15%/m, which is the operational criterion. Third, a PM10.0 value of 7,000 μg/m3 or less after CSD operation was proposed as a criterion for determining unwanted fire alarms. Through the verification test, it was also confirmed that the PM10.0 measurement was 7,000 μg/m3 or less or within 10% in all cases where ASD was 5%/m and ODM was 15%/m.
The English level of the paper needs to be improved. It is recommended to find a native English speaker for revision.
Author Response
Thank you for your review.
We totally agree with you. And we have revised the manuscript.
Since multiple files cannot be transferred from the system, we will send you the answer to the review as follows. Also, please note that the attached file is a revised manuscript. The cover-letter and picture files are provided by Ms. I mailed it to Kristy Weid. Please confirm. Thank you very much.
Response to Reviewer 4 Comments
Thank you for seeking assistance in improving the quality of our work. In response to reviewer’s comments, we have made amendments to the following documents.
Title Changes: Indoor Air Quality Sensor Utilization for Unwanted Fire Alarm Improvement in Studio-type Apartment
→ The title modified to better represent the purpose of the paper.
Point 1: In section 3.1, the ventilation of testbed site should be described in detail because it will influence the IAQS and PM value distribution. Is the experimental site chosen by the author representative?.
Response 1: Ventilation conditions were added. Thank you for your comment.
Point 2: The curves in Figures 10a,10b, 11a,11b, 15a and 16a cannot be seen clearly, it is recommended to redraw them.
Response 2: Thank you for improve the quality of the text. We modified each graph by dividing it into two parts. And as the structure of the paper has been revised, please note that the numbering of figures has also changed.

Round 2
Reviewer 2 Report
The authors have replied the former comments properly.
Reviewer 4 Report
The authors have revised paper carefully according to the review comments . I recommend that this paper can be published.
It is suggested to seek the assistance of an English professional to revise the paper writing.